# Physics-aware hand-object interaction denoising

## Abstract

The credibility and practicality of a reconstructed hand-object interaction sequence depend largely on its physical plausibility. However, due to high occlusions during hand-object interaction, physical plausibility remains a challenging criterion for purely vision-based tracking methods. To address this issue and enhance the results of existing hand trackers, this paper proposes a novel physically-aware hand motion de-noising method. Specifically, we introduce two learned loss terms that explicitly capture two crucial aspects of physical plausibility: grasp credibility and manipulation feasibility. These terms are used to train a physically-aware de-noising network. Qualitative and quantitative experiments demonstrate that our approach significantly improves both fine-grained physical plausibility and overall pose accuracy, surpassing current state-of-the-art de-noising methods.

## 1 Introduction

Hand pose tracking during hand-object interaction is a crucial task for various applications such as gaming, virtual reality, and robotics. Vision-based hand tracking methods have made significant progress in recent years by estimating hand poses from vision data sequences. However, heavy occlusions often occur during hand-object interaction, leading to ambiguity for vision-based trackers. Consequently, even state-of-the-art tracking methods still produce obvious errors and generate physically implausible artifacts such as inter-penetrating hand-objects and unrealistic manipulation. It is critical to remove such artifacts, increase the physical plausibility, and therefore ensure the usefulness of tracking results.

Several previous works have proposed to post-process the estimations generated by vision-based trackers using de-noising techniques. Some works have attempted to optimize hand poses by minimizing low-level penetration and attraction energy Chen et al. (2022). However, such optimization methods may struggle to handle severe noise, as they easily get stuck at local optima. Other works instead train neural networks for de-noising purposes Grady et al. (2021)Zhou et al. (2022), leveraging data-driven pose or contact priors to correct potentially significant pose errors. Nevertheless, purely data-driven approaches rely on the quality of the training dataset labels to achieve satisfactory visual effects and physical plausibility, and may be susceptible to overfitting and over-smoothing. Additionally, there is no guarantee that the resulting neural network is physically-aware.

In an effort to overcome the limitations of existing methods, we propose to combine data-driven de-noising with explicit modeling of the physical plausibility. In particular, such modeling needs to cover two essential aspects: i) grasp credibility, which demands that the hand pose in each frame be realistic given the object's geometry, avoiding interpenetration in particular; ii) manipulation feasibility, which considers the object's movement and requires proper hand-object contact that can plausibly explain the object's trajectory through hand manipulation, complying with physical laws.

Incorporating physics-based constraints into data-driven de-noising necessitates reshaping the loss landscape, enabling the network to learn to de-noise hand motions while adhering to physical constraints. While this idea appears straightforward, achieving it is difficult due to the intricate and non-differentiable process of verifying the plausibility of hand motions. Furthermore, when the de-noising algorithm violates physical constraints, it is necessary to provide a suitable path to guide it back to feasible motions. Meeting this requirement is even more challenging.

To address the aforementioned challenges, we introduce neural physical losses for assessing grasp credibility and manipulation feasibility, respectively. These losses are differentiable and can approximate non-differentiable and computationally intensive physical metrics effectively. Furthermore, they not only differentiate physically invalid hand motions from valid ones but also offer good pro-

jection directions to correct physically implausible hand motions. We integrate these neural physical losses into a novel hand motion denoising framework. Specifically, we design a de-noising auto-encoder that operates on a dual hand-object-interaction representation, along with a two-stage training process that effectively balances physical constraints and data priors.

To demonstrate the effectiveness of our designs, we conduct experiments on both data with synthetic errors and actual errors caused by trackers and achieve both qualitative and quantitative improvements over the previous state of the arts. To sum up, our main contributions are: **First**, we propose a physically-aware hand-object interaction de-noising framework which nicely combines data priors and physics priors. **Second**, we introduce differentiable neural physical losses to model both grasp credibility and manipulation feasibility to support end-to-end physically-aware de-noising. **Third**, we demonstrate the generalization of our neural physical losses regarding object, motion, and noise pattern variations, and show their effectiveness on different benchmarks.

## 2 RELATED WORK

**Hand reconstruction and tracking** The problem of reconstructing 3D hand surfaces from RGB or depth observations has garnered considerable attention in research. The existing body of work can be broadly classified into two distinct paradigms. Discriminative approaches focus on directly estimating hand shape and pose parameters from the observation, employing techniques such as 3D CNNs and volumetric representations Ge et al. (2019); Chen et al. (2021); Zhao et al. (2020); Malik et al. (2020); Boukhayma et al. (2019). In contrast, generative approaches adopt an iterative optimization process to refine a parametric hand model, iteratively aligning its projection with the observed data Sridhar et al. (2014); Taylor et al. (2016; 2017). While recent advancements have explored more challenging scenarios, such as reconstructing two interacting hands Zhang et al. (2021a); Mueller et al. (2019); Smith et al. (2020), these approaches often overlook the presence of objects, leading to decreased reliability in interaction-intensive scenarios. The absence of object-awareness limits their ability to accurately reconstruct hand surfaces in dynamic environments.

**Hand pose denoising** The goal of hand pose denoising is to improve the reliability and accuracy of the hand pose estimation or tracking system, enabling more robust and realistic hand motion analysis and interaction in applications such as virtual reality, augmented reality, robotics, and human-computer interaction. TOCH Zhou et al. (2022)improves motion refinement by establishing spatio-temporal correspondence between objects and hands. GraspTTA Jiang et al. (2021) utilizes contact consistency reasoning to generate realistic and stable human-like grasps. D-Grasp Christen et al. (2022) generates physically realistic and dynamic grasps for interactions between the hand and objects. Grabnet Taheri et al. (2020) aims to create accurate and visually realistic hand mesh models while interacting with previously unseen objects.

**Physical plausibility in hand-object interaction** The physical plausibility during hand-object interaction has been studied by many previous works. Zhang et al. (2021b) uses a neural network to learn from human motion and synthesize physics-plausible manipulation. Christen et al. (2022) and Yang et al. (2022) generate a physics-based hand control policy with deep reinforcement learning to reach specific grasping or moving goals. While these works focus on synthesizing hand motions, some works, such as Grady et al. (2021), and Hu et al. (2022), explore reconstructing or refining hand-object interaction leveraging physics priors, which is more relevant to our work. However, these works have limitations such as being limited to static grasps, relying on purely data-driven approaches and assuming only finger tips as source of forces.

## 3 METHOD

In this section, we describe our method for physically-aware hand pose de-noising. We begin by introducing the problem and outlining our approach. Given a potentially noisy hand pose trajectory during human-object interaction, represented by a sequence of hand meshes over $T$ frames denoted by $\widetilde{H} = (\widetilde{H}^i)_{1 \leq i \leq T}$ with $\widetilde{H}^i \in \mathbb{R}^{K \times 3}$, our method conditions on the object's geometry and dynamic information to refine the input and get more physically-plausible results $\hat{H} = (\hat{H}^i)_{1 \leq i \leq T}$. We consider hand-object interaction sequences containing a single hand and a single rigid object. Let $O = (O^i)_{1 \leq i \leq T}$ with $O_i \in \mathbb{R}^{L \times 3}$ denote object vertices over $T$ frames.

To combine data priors and physics priors in the de-noising process, we introduce a dual representation of hand-object interaction $F = (F^i)_{1 \leq i \leq T} = (\mathcal{F}(\hat{H}^i, O^i))_{1 \leq i \leq T}$ that enables physical

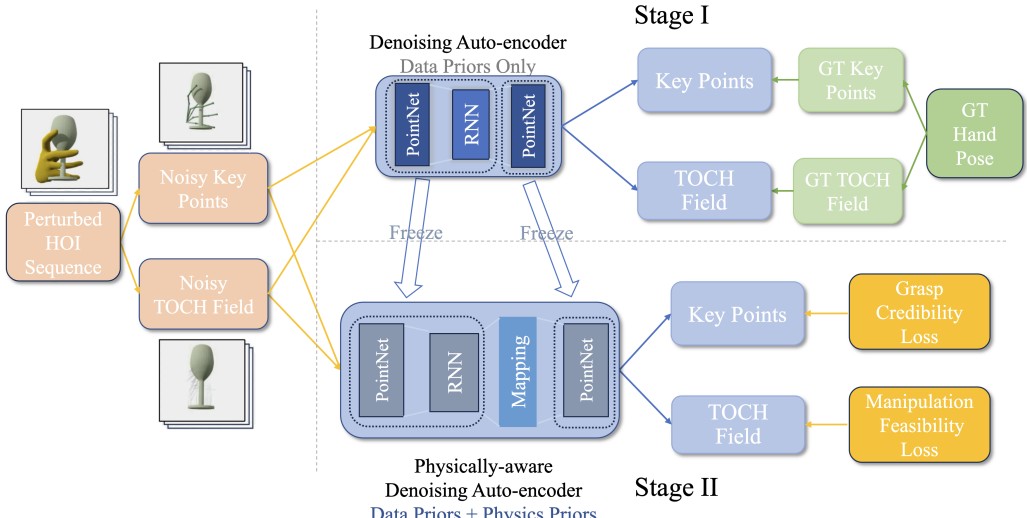

Figure 1: Overview of the training process.

reasoning besides capturing data priors about hand-object interaction. We elaborate on this representation in Section 3.1. Using $F$ as an intermediate representation, we refine hand poses via mapping noisy representation $\widetilde{F}$ to its correct version $\hat{F}$ with a de-noising auto-encoder. Using the refined representation $\hat{F}$, the corrected hand pose sequence $\hat{H}$ is fitted. Details regarding the architecture of our de-noising network can be found in Section 3.1.

To produce more physically plausible results, during the training of the de-noising network, besides traditional data losses, we propose two neural physical loss terms for assessing grasp credibility and manipulation feasibility. They are capable of conducting physical reasoning given the HOI representation $F$ and producing assessment scores differentiable to $F$. What's more, with carefully designed training scheme and training targets, these loss terms form smooth landscape, and therefore yield contributive signals that effectively guide the de-noising network to gain physical-awareness and produce more physically-plausible results. Section 3.2 and 3.3 describe the construction of these two loss terms. And we describe their usage in the training process in Section 3.1.

## 3.1 TRAINING FRAMEWORK

**Dual HOI representation**   Instead of directly using hand vertices as the representation in the de-noising process, we propose a dual hand-object interaction representation that combines holistic hand pose information and fine-grained hand-object relation. This intermediate representation bridges explicit hand mesh vertices and physics-related knowledge learned by the proposed neural loss terms, allowing knowledge to flow between them and eventually improve the physical plausibility of the refined hand pose. On the one hand, by emphasizing hand-object relation at two different granularity, this dual representation enables the physical loss terms to reason about contact and deep penetration cases and better evaluate the physical plausibility of a given hand pose. On the other hand, with its focus on the physics-related characters of hand poses, when used to fit the explicit hand pose representation, this proposed representation can effectively improve vital aspects of the hand pose regarding physical plausibility.

Given hand vertices $H = (H^i)_{1 \leq i \leq T}$ of hand MANO Romero et al. (2022) meshes and object vertices $O = (O^i)_{1 \leq i \leq T}$ over $T$ frames. The whole HOI representation $(F^i)_{1 \leq i \leq T}$ we use is denoted as:

$$F^i = (S^i, \mathcal{T}^i)$$

We regress the 21 hand key points $(S^i)_{1 \leq i \leq T}$ with $S^i \in \mathbb{R}^{21 \times 3}$ from hand vertices and model object-centric hand-object correspondence with the implicit field proposed by Zhou et al. (2022), that is, $(\mathcal{T}^i)_{1 \leq i \leq T}$ where $\mathcal{T}^i = \{\mathcal{C}^i_j\}^N_{j=1} = \{(m^i_j, d^i_j, p^i_j)\}^N_{j=1}$. With $N$ points randomly sampled on the object surface in the $i$-th frame, $\mathcal{C}^i_j$ represents the corresponding hand point of the $j$-th object point. To find hand-object correspondence, we cast rays in the object surface's normal directions from the sampled points and consider hand points the rays first hit, as the corresponding points. $m^i_j$ is set to 1 if the $j$-th object point has corresponding hand point and is 0 otherwise. $d^i_j$ denotes the distance

between the $j$-th object points and its corresponding hand point, while $p_j^i \in \mathbb{R}^3$ encodes the semantic information of the hand point encoded as its position in the canonical space. After obtaining the refined representation, we can fit hand mesh to the representation and obtain the denoised hand poses.

**De-noising anto-encoder** We use an auto-encoder as our backbone, which consumes noisy data in the form of our dual HOI representation $\widetilde{F}$ and produces the corrected version $\hat{F}$. Our de-noising network adopts the PointNet Qi et al. (2017) architecture to process the the TOCH field part $\mathcal{T}^i$ of the input representation into a global object feature for the $i$-th frame, which is then concatenated with the hand-centric part $S^i$ in our representation to form the HOI frame feature $x^i$. $(x^i)_{1 \le i \le T}$ is considered as time series data and further processed by a RNN, whose output is used to decode the refined representation $\hat{F}$.

**Training process** To obtain a physically aware de-noising network, we train the auto-encoder in two stages as shown in Figure 1. In the stage I, we train the model with supervision from ground truth data, learning a projection mapping from noisy hand pose representation to the clean data manifold. The training loss of stage I can be expressed as:

$$\mathcal{L}_{\mathrm{I}} = \alpha_{\mathcal{T}} \mathcal{L}_{\mathcal{T}}(\hat{\mathcal{T}}, \mathcal{T}_{\mathrm{GT}}) + \alpha_S \mathcal{L}_S(\hat{S}, S_{\mathrm{GT}})$$

where $\mathcal{L}_{\mathcal{T}}$ and $\mathcal{L}_S$ measure the difference between the refined representation $\hat{F}$ and the ground truth representation $F_{GT}$. To reinforce the physical awareness of the de-noising network, we introduce the two physical loss terms in stage II. Specifically, we freeze the auto-encoder trained in stage I, but plug in a mapping layer $\phi$ between the encoder and the decoder, which is trained with the neural physical losses in stage II. $\phi$ consumes latent vectors produced by the encoder, together with the linear accelerations of the object obtained with finite difference method. As the dual representation is in object-frame, it is necessary to introduce of linear acceleration to enable $\phi$ to reason about manipulation feasibility. We enable $\phi$ to capture the physical reasoning enforced by the two losses, therefore reshaping the clean hand pose manifold learned in stage I.The training loss of stage II integrates three components:

$$\mathcal{L}_{II} = \alpha_{\mathrm{grasp}} \mathcal{L}_{\mathrm{grasp}}(\hat{S}, O) + \alpha_{\mathrm{manip}} \mathcal{L}_{\mathrm{manip}}(\hat{\mathcal{T}}) + \alpha_{\mathrm{reg}} ||\phi(v, a) - v||_2$$

where $\mathcal{L}_{\mathrm{grasp}}$ and $\mathcal{L}_{\mathrm{manip}}$ are the learned physical losses to be explained in the following sections, while $v$ denotes the latent vector produced by the encoder and $a = (a^i)_{1 \le i \le T}$ denotes the linear accelerations of the object. The regulation term ensures proximity between the mapped and the original latent vector, therefore maintaining the data priors learned in the first stage.

## 3.2 GRASP CREDIBILITY LOSS

Given a single frame from a HOI sequence, with no knowledge about the movement of the object, the geometry relation between hand and object is the most important clue for humans to evaluate the plausibility of such a frame. To be specific, the hand pose must conform to the object's geometry, forming a plausible grasp and avoiding penetration or collision. While many recent works, such as Hasson et al. (2021), Gundogdu et al. (2019) and Buffet et al. (2019), focus on mitigating hand-object penetration during hand pose estimation, they tend to be helpless when dealing with deep penetration cases where the hand penetrates through the object completely instead of just entering the object surface, which are common when dealing with thin and delicate object parts.

When deep penetration happens, the direction to move the vertices to resolve penetration becomes ambiguous as the hand mesh lies on both side of the object, and it is very challenging to have a differentiable term producing correct gradients. While our method understands how to handle deep penetration through data prior learned from a novel penetration depth metric $PD$.

$PD$ quantifies the severity of penetration of a noisy hand pose $\widetilde{H}^i$, which is paired with the corresponding clean hand pose $H^i$, with respect to the object mesh vertices $O^i$. The metric is computed by comparing $\widetilde{H}$ with the ground truth hand mesh vertices.

To compute this penetration depth metric, we focus on the hand vertices in $\widetilde{H}$ whose counterparts in $H$ are in contact with the object. Specifically, we consider hand points with distances less than a threshold $c_{\mathrm{contact}} = 2\mathrm{mm}$ to the object surface as contact points. For $C$ contact points $\{h_i\}_{i=1}^C$ on the ground truth hand mesh, whose counterparts on the evaluated mesh are denoted as $\{\widetilde{h}_i\}_{i=1}^C$, we compute their shift between two meshes as $\{\vec{h}_i = \widetilde{h}_i - h_i\}_{i=1}^C$. At the object contact points $\{o_i\}_{i=1}^C$,

Figure 2: The proposed grasp credibility loss and manipulation feasibility loss can help quantify the physical plausibility of hand pose estimation results in a smooth way. Curves on the left depict the loss landscape of neural losses trained with different target. In the images on the right, the darkness of hand colors indicates the value of our proposed neural losses. Notice that loss trained with hard target only (Loss-) produce less smooth landscape.

we compute the normal $\{\vec{n}_i\}_{i=1}^C$ of the object surface. Then the metric is computed as:

$$PD(\widetilde{H}^i, H^i \mid O^i) = (\max_{\substack{j=1,2,\ldots,C \\ ||\vec{h}_j-(\vec{n}_j\cdot\vec{h}_j)\vec{n}_j||<c_{\text{tangent}}}} (-\vec{n}_j \cdot \vec{h}_j))^+$$

where $x^+$ denotes $\max(0,x)$ and $c_{\text{tangent}}$ is set to 1cm empirically so that only hands vertices with small shift in directions orthogonal to the object surfaces are considered for penetration evaluation, avoiding false positive cases where the hand vertices shift so much that the their positions with respect to the object change completely. This definition can be considered as an approximation of the penetration depth at the most severe penetration position. Compared with previous works that measure the intersection volume or count hand vertices inside the object mesh to evaluate hand-object inter-penetration, our metric better reflects deep penetration cases.

This metric, however, isn't differentiable to our intermediate hand-object interaction representation, hence can't be directly used for improving the geometry credibility of results produced by our denoising network. We train a physics-aware neural network $\psi_{\text{grasp}}$ with a PointNet-like backbone that consumes hand skeletons and object point clouds to produce prediction results $p$ between 0 and 1, where 0 indicates no severe penetration and 1 indicates the opposite. We introduce a threshold $c_{\text{PD}}$ use the comparison result of $PD > c_{\text{PD}}$ as a hard target, as well as the original $PD$ as a soft training target to encourage smooth prediction output. The loss for training $\mathcal{L}_{\text{grasp}}$ can be defined as:

$$\mathcal{L}_{\text{grasp train}}(p, PD) = \alpha_{\text{grasp hard}}\text{BCE}(p, b_{\text{hard}}) + \alpha_{\text{grasp soft}}\text{BCE}(p, b_{\text{soft}})$$

$$b_{\text{hard}} = \mathbf{1}_{PD \geq c_{\text{PD}}}, \quad b_{\text{soft}} = 1 - e^{-c_{\text{soft}}*PD}$$

We set $c_{\text{soft}} = \frac{ln(2)}{c_{\text{PD}}}$ such that $b_{\text{soft}} = 0.5$ when $PD = c_{\text{PD}}$ to make sure that $b_{\text{hard}}$ and $b_{\text{soft}}$ are consistent. We set $c_{\text{PD}} = 1.5$cm empirically. $\text{BCE}(\cdot, \cdot)$ denotes the binary cross entropy function.

As shown in Figure 2, the combination of soft target and hard target during the training allows $\mathcal{L}_{\text{grasp}}$ to distinguish deep penetration cases smoothly, improving its usability as a loss term.

We also attempt to improve the generalization ability of $\mathcal{L}_{\text{grasp}}$ and the loss landscape smoothness by assuring data variation in its training dataset. Using a HOI dataset containing ground truth data $\{(H^i, O^i)\}_{i=1}^D$, we first get its perturbed version $\{(\widetilde{H}^i, O^i)\}_{i=1}^D$ by adding Gaussian noise to the MANO parameters. To assure variation in noise magnitude, we further conduct linear interpolation between the MANO parameters of the ground truth data and the perturbed data to get $m$ hand poses from each perturbed-clean hand pose pair. The dataset we finally obtain can be expressed $\bigcup_{i=1,2,\ldots,D}\{(\breve{H}_j^i, O^i)\}_{j=1}^m$ where $\breve{H}_j^i$ denotes the $j$-th interpolation result between $\breve{H}_1^i = H^i$ and $\breve{H}_m^i = \widetilde{H}^i$.

### 3.3 MANIPULATION FEASIBILITY LOSS

For a HOI sequence to be realistic, besides grasp credibility that only focuses on the single frame, whether the object's movement seems feasible also matters. To this end, we propose two manipulation feasibility metrics, force error ($FE$) and manipulation expense ($ME$), that evaluate whether the given hand-object contact can feasibly move the object along its trajectory. The two metrics are used as hard target and soft target respectively to train the neural physical loss term $\mathcal{L}_{\mathrm{manip}}$. The two metrics, as well as the neural loss term $\mathcal{L}_{\mathrm{manip}}$, take the hand-object correspondence part $\mathcal{T}$ of our representation and the object's linear acceleration $a$ as input.

In the first force error metric, we measure to what extent can the object's movement be explained by forces applied at the the contacts within the corresponding friction cones.

Given an implicit field $\mathcal{T}^i$, we consider the $M$ contact points among the randomly sampled $N$ points on the object surface, and denote the object surface normal at these contact points as $\{\vec{n}_j\}_{j=1}^M$. Let $\vec{F} = m_0(-\vec{g} + \vec{a})$ denote the force needed for the object with mass $m_0$ to achieve its acceleration $\vec{a}$ when subject to gravity $m_0\vec{g}$. We obtain $\vec{a}$ with finite difference method. And let $\{\vec{f}_j\}_{j=1}^M$ with $f_j \in \mathbb{R}^3$ denotes the set of contact forces applied to the object that we solve for. We require the forces to lie in the corresponding friction cones specified by the coefficient of static friction $\mu$. Then the force error metric can be expressed as:

$$FE(\{\vec{n}_j\}_{j=1}^M, \vec{F}) = \min_{\frac{\vec{f}_j}{||\vec{f}_j||} \cdot (-\vec{n}_j) \geq \sqrt{\frac{1}{1+\mu^2}}} ||\sum_{j=1}^M \vec{f}_j - \vec{F}||$$

Notice that $FE$ is always between 0 and 1, and ideally, for a physically plausible frame, $FE$ should be 0. Since we only care about the relative force error, $m_0$ can be set to any non-zero constant. We set $m_0 = 1$kg. $\mu$ is set to 0.8 empirically. We elaborate on the choices of these physical properties in the supplementary material.

While this force error metric can be used as a binary result to verify whether the movement of the object is feasible given a certain hand pose, a major drawback is that its value doesn't correctly reflect how infeasible a hand pose is. However, to obtain a loss term with smoother landscape, a metric with continuous result indicating the degree of manipulation feasibility would be more favorable. Therefore, we propose the manipulation expense metric that evaluates how far the given hand pose is from the closest feasible hand pose.

Intuitively, this manipulation expense metric considers all the plausible force distribution maps that yield the required total force, and find the one that best match the current contact map, in that least forces are applied at object points which are actually far from the hand. The difference between the current contact map and its best match found in the above manner can reflect the quality of the current contact map regarding manipulation feasibility.

In this metric, we consider all $N$ sampled object points in $\mathcal{T}^i$, and $d_j$ denotes the signed distance between the $j$-th sampled object point and its corresponding hand point. Let $\{\vec{f}_j\}_{j=1}^N$ with $\vec{f}_j \in \mathbb{R}^3$ denote potential forces exerted at the $N$ sampled points, the manipulation expense metric can be expressed as:

$$ME(\{\vec{n}_j\}_{j=1}^N, \{d_j\}_{j=1}^N, \vec{F}) = \min_{\substack{\frac{\vec{f}_j}{||\vec{f}_j||} \cdot (-\vec{n}_j) \geq \sqrt{\frac{1}{1+\mu^2}} \\ \sum_{j=1}^N \vec{f}_j = \vec{F}}} \sum_{j=1}^N ||\vec{f}_j|| \cdot (|d_j| - c_{\mathrm{contact}})^+$$

We calculate $FE$ and $ME$ by solving for $\{\vec{f}_j\}_{j=1}^M$ and $\{\vec{f}_j\}_{j=1}^N$ through optimization processes respectively. Please refer to our supplementary material for details. We train a PointNet-like neural predictor $\psi_{\mathrm{manip}}$ that produces a output $q$ between 0 and 1, to form the manipulation feasibility loss. The training loss of $\psi_{\mathrm{manip}}$ is formed as:

$$\mathcal{L}_{\mathrm{manip\_train}}(q, FE, ME) = \alpha_{\mathrm{manip\ hard}} \mathrm{BCE}(q, s_{\mathrm{hard}}) + \alpha_{\mathrm{manip\ soft}} \mathrm{BCE}(q, s_{\mathrm{soft}})$$

$$s_{\mathrm{hard}} = \mathbf{1}_{FE \geq c_{FE}}, \quad s_{\mathrm{soft}} = \mathbf{1}_{FE \geq c_{FE}} \cdot (0.5 + \frac{arctan(ME)}{\pi})$$

$\alpha_{\mathrm{manip\ hard}}$ and $\alpha_{\mathrm{manip\ soft}}$ are constant weights. We use the same training dataset for $\mathcal{L}_{\mathrm{manip}}$ as the one used to train $\mathcal{L}_{grasp}$.

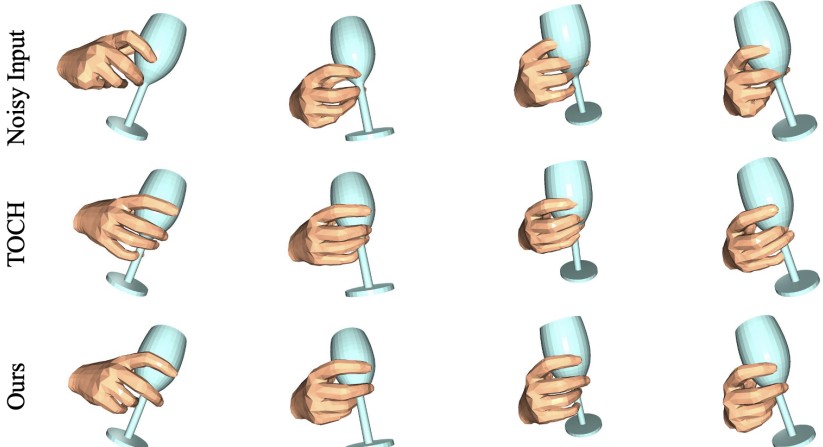

Figure 3: Qualitative results on GRAB dataset. We can see that TOCH produces physically implausible results when dealing with thin and delicate parts of objects, while our results are more realistic.

# 4 EXPERIMENTS

In this section we evaluate the proposed method on data with synthetic noise and actual tracking noise. We first introduce the datatsets (Section 4.1) and the evaluation metrics (Section 4.2). Results of our approach on correcting synthetic tracking error and refining results from vision-based trackers are in (Section 4.3) and (Section 4.4) respectively. Finally, we conduct ablation studies to evaluate the advantages of our physically-aware loss design in Section 4.5.

## 4.1 DATASETS

**GRAB**. We train our de-noising network and the two neural loss terms on GRAB Taheri et al. (2020), a MoCap dataset for whole-body grasping of objects containing 51 objects from Brahmbhatt et al. (2020). We follow the recommended split and select 10 objects for evaluation and testing.

**HO-3D**. HO-3D is a dataset of hand-object interaction videos captured by RGB-D cameras, paired with frame-wise annotations of 3D hand poses and object poses. We use HO-3D to evaluate how well our pipeline trained with synthetic pose errors can generalize to real tracking errors produced by vision-based trackers. We use the second official release of HO-3D.

## 4.2 METRICS

**Mean Per-Joint Position Error (MPJPE)** and **Mean Per-Vertex Position Error (MPVPE)**. We report the average Euclidean distances between refined and ground truth 3D hand joints and vertices. These metrics measure the accuracy of hand poses and shapes.

**Intersection Volume (IV)**. This metric measures volume of the intersection between the hand mesh and the object mesh. It reflects the degree of hand-object inter-penetration.

**Penetration Depth (PD)**. To better measure cases of deep penetration, we also report the penetration depth metric proposed in Section 3.2. Since the ground truth hand poses of HO-3D are withheld, we cannot calculate PD on HO-3D and only report it on GRAB.

**Contact IoU**. This metric assesses the Intersection-over-Union between the ground truth contact map and the predicted contact map. The contact maps are obtained by considering object vertices within $2\,mm$ from the hand as in contact. This metric is also reported on GRAB only.

**Plausible Rate**. To consider both grasp credibility and manipulation feasibility for a holistic assessment regarding physical plausibility, we use this metric that reflects both aspect. For a given frame of hand pose to be plausible, i) the PD metric should be less than $1.5\mathrm{cm}$ (this threshold is chosen following Zhang et al. (2021b)), ii) the force error metric proposed in 3.3 should be less than $0.1$. Since PD is not available for HO-3D, we only consider ii) for evaluation on HO-3D.

## 4.3 REFINING SYTHETIC ERROR

To employ our de-noising method in more realistic applications and achieve best potential performance, it would be ideal to train the model on the predictions of the target tracking method to be augmented. Yet this might lead to overfitting to tracker-specific noise, which is undesirable for generalization. Hence we trained the network on dataset with Gaussian noise of different composition and magnitude, which synthesizes the hand errors that tracking methods induce. Quantitative results are shown in Table 1 and qualitative results are presented in Figure 3

Table 1: Quantitative results on refining GRAB dataset with sythetic noise. T-0.01 denotes the dataset with translation noise complying the distribution $\mathcal{N}(0, 0.01)$, $\theta$-0.3 denotes the dataset with pose noise complying the distribution $\mathcal{N}(0, 0.3)$, other noise patterns are denoted similarly. We additionally add $\Delta r \sim \mathcal{N}(0, 0.05)$ to the global orientation in all the datasets. We train the network only on the "T-0.01, $\theta$-0.3" dataset and test it on datasets with different noise patterns. Our method is robust to different noise patterns and magnitudes, especially in its ability to produce physically plausible results.

| | T-0.01 | T-0.02 | $\theta$-0.3 | $\theta$-0.5 | T-0.01 $\theta$-0.3 | T-0.02 $\theta$-0.5 |
|---|---|---|---|---|---|---|
| MPJPE | $16.01 \rightarrow \mathbf{6.91}$ | $27.83 \rightarrow \mathbf{10.01}$ | $7.52 \rightarrow \mathbf{5.61}$ | $8.74 \rightarrow \mathbf{6.38}$ | $18.86 \rightarrow \mathbf{7.39}$ | $31.49 \rightarrow \mathbf{11.97}$ |
| MPVPE | $16.32 \rightarrow \mathbf{6.24}$ | $28.43 \rightarrow \mathbf{10.46}$ | $\mathbf{6.23} \rightarrow 6.31$ | $8.85 \rightarrow \mathbf{6.97}$ | $19.13 \rightarrow \mathbf{6.78}$ | $33.25 \rightarrow \mathbf{11.24}$ |
| contact IoU | $3.31 \rightarrow \mathbf{24.82}$ | $2.39 \rightarrow \mathbf{21.77}$ | $5.45 \rightarrow \mathbf{25.14}$ | $4.46 \rightarrow \mathbf{24.18}$ | $3.62 \rightarrow \mathbf{23.94}$ | $2.47 \rightarrow \mathbf{20.50}$ |
| IV | $0.91 \rightarrow \mathbf{0.90}$ | $1.43 \rightarrow \mathbf{1.10}$ | $\mathbf{0.88} \rightarrow 0.92$ | $1.77 \rightarrow \mathbf{0.94}$ | $\mathbf{0.87} \rightarrow 1.13$ | $2.35 \rightarrow \mathbf{1.12}$ |
| PD | $0.77 \rightarrow \mathbf{0.32}$ | $0.85 \rightarrow \mathbf{0.43}$ | $0.74 \rightarrow \mathbf{0.44}$ | $1.22 \rightarrow \mathbf{0.47}$ | $0.80 \rightarrow \mathbf{0.44}$ | $1.56 \rightarrow \mathbf{0.45}$ |
| plausible rate | $0.42 \rightarrow \mathbf{0.95}$ | $0.33 \rightarrow \mathbf{0.92}$ | $0.45 \rightarrow \mathbf{0.93}$ | $0.43 \rightarrow \mathbf{0.93}$ | $0.42 \rightarrow \mathbf{0.91}$ | $0.31 \rightarrow \mathbf{0.90}$ |

Table 2: Quantitative results on HO-3D dataset. The hand joint and mesh errors are obtained after Procrustes alignment following the official evaluation protocol of HO-3D.

| | MPJPE | MPVPE | IV | plausible rate |
|---|---|---|---|---|
| Hasson *et al.* | 11.4 | 11.4 | 8.75 | 0.71 |
| TOCH | 10.9 | 11.3 | 7.24 | 0.73 |
| Ours | 10.7 | 11.2 | 5.95 | 0.85 |

## 4.4 REFINING VISION-BASED HAND TRACKER

We also use our network to refine the results produced by state-of-the-art vision-based models on HO-3D dataset. To evaluate how well our method generalize to actual tracker error, both the de-noising auto-encoder and the two neural loss terms are trained only on GRAB with synthetic errors, and then used on HO-3D without further fine-tuning. Hasson *et al.* Hasson et al. (2020), a RGB-based hand pose tracker is used to predict hand poses on the test split of HO-3D dataset. Zhou et al. (2022) and our method are used to refine the results it produces. The results are shown in Table 2. Physical plausibility is significantly improved, which is indicated by IV and plausibility rate.

## 4.5 ABLATION STUDIES

**Physically-aware loss**. To demonstrate the advantages of our proposed physically-aware loss terms in modeling and improving physical plausibility, in stage II, we remove signals from the two neural loss terms perspectively and train two baseline de-noising networks. Comparison between them and our complete method is presented in Table 3.

Without the grasp credibility loss, the network tends to more actively adhere the hand to the object surface, increasing the contact area so that FE and ME can likely be reduced. However, this is achieved at the expense of more severe hand-object inter-penetration, indicated by IV and PD. Removing the manipulation feasibility loss induces the opposite result. While lower IV and PD are attained, cases where the object is hovering in the air appear more frequently.

**Soft target for training neural losses**. When training the physically-aware neural loss terms, we use both soft target and hard target for smooth loss landscape. To demonstrate the advantages of this design, we train baseline neural losses with hard target only and compare them with neural losses trained with both targets, and present comparisons in Table 4 and Figure 2 regarding their classification performance on test set and ability of improving the de-noising network when exploited.

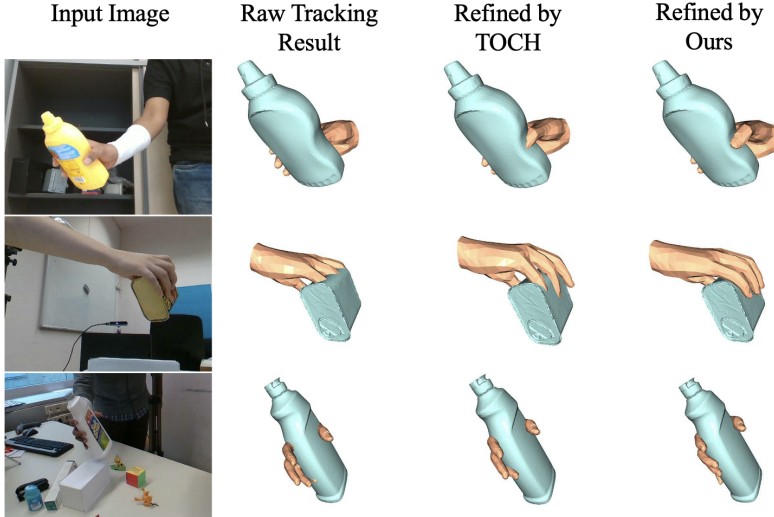

| Input Image | Raw Tracking Result | Refined by TOCH | Refined by Ours |

Figure 4: Qualitative results on HO-3D dataset. Our method effectively denoises tracking result, and produces more physically plausible hand-object interaction than TOCH.

Table 3: Ablation on the physical neural losses.

|  | MPJPE | MPVPE | contactIoU | IV | PD | plausible rate |
|---|---|---|---|---|---|---|
| GT | - | - | - | 0.75 | - | 0.92 |
| TOCH | 13.17 | 12.24 | 21.83 | 2.24 | 0.55 | 0.78 |
| Ours(w/o $\mathcal{L}_{manip}$) | 7.41 | 6.78 | 23.96 | 0.96 | 0.41 | 0.82 |
| Ours(w/o $\mathcal{L}_{grasp}$) | 7.28 | 6.65 | 23.51 | 2.43 | 0.59 | 0.81 |
| Ours | 7.39 | 6.78 | 23.94 | 1.13 | 0.44 | 0.91 |

Table 4: We use the losses to discern implausible frames on the test split of Perturbed GRAB and report their F-scores. While the classification performance of losses trained with two types of target choices are close, neural losses trained with both targets yield better results when exploited.

(a) Target used to train grasp credibility loss.

|  | F-score | PD | IV |
|---|---|---|---|
| hard | 0.85 | 0.49 | 1.52 |
| hard + soft | 0.93 | 0.44 | 1.13 |

(b) Target used to train manipulation feasibility loss.

|  | F-score | plausible rate |
|---|---|---|
| hard | 0.90 | 0.84 |
| hard + soft | 0.88 | 0.91 |

## 5  CONCLUSION

We propose a physically-aware hand-object interaction de-noising framework which combines data priors and physics priors to generate plausible results.In particular, our differentiable neural physical losses effectively assess grasp credibility and manipulation feasibility of given hand poses and form smooth loss landscape, enabling physically-aware de-noising. Experiments demonstrate that our method generalize well to novel objects, motions and noise patterns.

## REPRODUCIBILITY STATEMENT

We have provided implementation details in the main paper and supplementary material. We will release the code to reproduce our results upon acceptance.

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
