# OpenReview forum: "Physics-aware Hand Object Interaction Denoising"
_ICLR.cc/2024/Conference — ICLR 2024 Conference Withdrawn Submission_

### Official Review · Reviewer_CERf · 2023-10-29

**Soundness:** 2 fair
**Presentation:** 2 fair
**Contribution:** 2 fair
**Rating:** 3
**Confidence:** 4

**Summary:**

This paper proposes a physically-aware hand motion de-noising method to address the challenge of physical plausibility in hand-object interaction tracking. The method introduces two loss terms that capture grasp credibility and manipulation feasibility, which are used to train a physically-aware de-noising network. Experiments show that this approach outperforming other de-noising methods.

**Strengths:**

This paper proposes a dual hand-object interaction (HOI) representation for training a model that improves the physical plausibility of refined hand poses. The proposed representation combines holistic hand pose information and fine-grained hand-object relation, bridging explicit hand mesh vertices and physics-related knowledge learned by neural loss terms.

**Weaknesses:**

* The presence of numerous empirical settings in the two losses is observed, accompanied by a noticeable absence of a well-defined physical basis.

* It should be noted that certain significant references are missing, such as the reference for the HO-3D dataset. Red color and blue color are not defined in the caption or in the main text.

* The degree of novelty exhibited by this paper appears to be limited. The sole introduction of two loss terms, resulting in marginal improvements in MPJPE, constitutes the primary contribution of this work. Penetration depth metric is also used in previous papers.

**Questions:**

* How to balance the data prior and physical constraint in the two-stage training process?

---

### Official Review · Reviewer_Xkao · 2023-10-31

**Soundness:** 2 fair
**Presentation:** 2 fair
**Contribution:** 2 fair
**Rating:** 3
**Confidence:** 4

**Summary:**

Given a sequence of noisy hand poses, this paper aims to recover a noise-free and physically plausible hand motion. To this end, the authors propose a two stage pipeline with the first stage is a denoising auto-encoder trained with GT hand pose similar to that from TOCH (Zhou et al. (2022)). The second stage aims to refine the results with two novel physics-aware loss: grasp credibility loss and manipulation feasibility loss. The former aims to sloves the hand object penetration problem. And the latter measures ''whether the given hand-object contact can feasibly move the object along its trajectory''. The paper provides experimental results on synthetic dataset created by adding Gaussian noise to GT hand pose and test the generalization on one real dataset.

**Strengths:**

The motivation of leveraging the force analysis to account for the physically feasibility of the HOI is very interesting. The proposed two losses are also novel.

**Weaknesses:**

- The writing of the paper is not very clear.
   - Section 3.1 second paragraph, the first sentence is not a valid sentence.
   - Section 3, the first paragraph, about the notation in $\tilde{H}=(\tilde{H}^{i})_{1\leq i \leq T}$, There are two issues here. Firstly, maybe I am wrong, but I do not think such definition of using brackets $( )$ are mathematically meaningful. Is it to define a set, if so it should use { }.  Secondly, ${1\leq i \leq T}$ defines a continuous variable from 1 to T not a discrete one.
  - Section 3.2, page 5 second paragraph, according to the context, a physics-aware neural network is trained to mimic the penetration depth metric (PD). After it is trained, how is it used? I believe it has been used to train the second stage, right? The same for section 3.3.
  - It seems all the inline references used in the paper are not in parenthesis. I believe the authors miss used \citet when there should be \citep.

- I am not convinced by the setup of the task. From my understanding, it only refines the hand pose regardless of the object pose. However, the object poses produced by the RGB based methods such as the one used in this paper (Hasson et al. (2020)), are also very noisy. The two proposed losses highly rely on the object poses. Will the noisy object pose affects the final results?
    - Given a noisy hand motion, should the denoising process be deterministic especially when the noise level is high?

- In tuitively, it is hard to understand the proposed losses especially for the manipulation expense metric (page 6).

- As to the experiments, it would be better to also evaluate on other real-world datasets like DexYCB with more recent hand-object pose estimation pipelines to predict the inital hand-object poses.

**Questions:**

Please see the weakness section

---

### Official Review · Reviewer_R2Kj · 2023-10-31

**Soundness:** 3 good
**Presentation:** 2 fair
**Contribution:** 2 fair
**Rating:** 6
**Confidence:** 4

**Summary:**

Authors proposed the physically-aware hand-object interaction de-noising framework combining data and physics priors. To support the framework, differentiable neural physical losses are provided to model both grasp credibility and manipulation feasibility. Also, the proposed method generalizes well to novel objects, motions, and noise patterns. In experiments, GRAB and HO-3D datasets are used to prove the superior performance.

**Strengths:**

It seems interesting to develop the denoising framework considering the physically-aware hand object interaction
New loss functions based on grasp credibility and manipulation feasibility were proposed.

**Weaknesses:**

The quality of Figure 2 could be improved. For example, it is nor clear what the frame in Fig 2 means (between training epoch or video’s frame). Also, it is unclear the right side of Fig. 2 is denoting the frames in videos or changes obtained during the optimization process.
Not clear which dataset was used in Table 3.
Technical novelty looks rather weak. Only few loss functions are proposed.
The accuracy gap to TOCH looks rather impressive; while the results presented in the ablation study is confusing. While the loss is the main contribution of the paper; the results obtained are not supporting the importance of them.
Should additionally show the ablation study on the HO-3D

**Questions:**

How d and p are defined when m is equal to 0? (In the case when there is no hand point corresponding to the object points).
In Table 3 and Table 4(b), when using losses, the accuracy seems gradually reduced. What is the reason for this?
What is the exact structure of the mapping network?

---

### Official Review · Reviewer_ZPXX · 2023-11-01

**Soundness:** 3 good
**Presentation:** 3 good
**Contribution:** 3 good
**Rating:** 8
**Confidence:** 3

**Summary:**

This paper proposes a method to denoise hand-object interaction sequences in a physically plausible manner. The proposed method follows a denoising autoencoder approach and employs two physically-aware loss terms to enforce physical plausibility while minimally perturbing the input poses. Experimental results demonstrate the effectiveness of the proposed approach in two different hand-object interaction datasets, including a cross-dataset experiment that exhibits good generalization.

**Strengths:**

This is a solid work that tackles a challenging problem. Clearly a lot of work has been invested towards achieving the presented results. The problem is inherently hard given its discrete nature. This is mentioned in the introduction in the sentence reading "due to the intricate and
non-differentiable process of verifying the plausibility of hand motions". Taking this argument one step further, one can argue that it's the discrete nature of the hand-object interaction (contact vs non-contact) that imposes the difficulty on the verification process. This is also the reason others have relaxed the problem by considering small proximity fields near the contacts and similar relaxation strategies.

Further strong points include the generalization obtained across datasets and the wide applicability of the tackled problem.

**Weaknesses:**

Despite being rather well written overall, there are some points that in my opinion are hard to follow, and would benefit from a restructuring/rewriting. Specifically, in my opinion the whole paragraph describing the intermediate representation should be restructured for readability, in its current state it is hard to follow. I realize that this mostly describes the implicit field representation by Zhou et al, but it would still be beneficial for this work to adequately describe it since it forms a core part of the employed representation. Furthermore, but not as importantly, you may also want to consider reordering the subsection order in the methodology Section. This will allow to first present the two proposed losses, and then use them to describe the training process, this might aid the overall readability of the section.

Consider stating fairly early in the paper (even perhaps in the abstract) that a *rigid* object is assumed -- this is of course fine, the problem is already hard enough even before considering deformable objects. Nevertheless I believe it would be beneficial for the interested reader that this assumption is explicitly stated early on.

Another criticism point arises from multiple empirical values used throughout the work, such as the object mass (although I realize this is essentially up to a scale), the friction coefficient, and several thresholds representing physical distances such as the contact threshold. Although the values seem reasonable, it might make sense to meta-optimize them or otherwise justify them in a more consistent manner.

To the best of my understanding, the intermediate mapping layer \phi is not explicitly described, apart from describing it as a "projection layer". Is this a single linear layer? Is a non-linearity used? Perhaps an MLP of short depth? Please clarify

Regarding the penetration depth metric, the paragraph defining it concludes "Compared with previous works .... , our metric better reflects deep penetration cases". Intuitively, how is this achieved? Are there any experiments to back this claim?

Another point that requires some attention is the perturbation strategy of the MANO parameters in the end of Section 3.2. This may need some attention, depending on what parameters are perturbed. MANO consists of pose and shape parameters, which may be further processed using PCA to reduce their total number. Where exactly are you applying the perturbation? How did you determine the (maximum) noise variance?

Finally, the tackled problem has a long history and is still very actively researched. Space permitting, consider also citing the following related works:
Oikonomidis et al. Full dof tracking of a hand interacting with an object by modeling occlusions and physical constraints
Kyriazis et al. Physically Plausible 3D Scene Tracking: The Single Actor Hypothesis
Brahmbhatt et al. ContactDB: Analyzing and Predicting Grasp Contact via Thermal Imaging (although you do cite and use the subsequent ContactPose, it would initiate the unfamiliar reader to this important line of work)

**Questions:**

Please feel free to address the points raised above

---

### Author Response · Authors · 2023-11-18

I would like to express my gratitude for the time and effort you have invested in reviewing our manuscript. Your comments and suggestions have been helpful in highlighting areas that require clarification and improvement. While we have carefully considered each point of feedback and recognize the potential enhancements they suggest for our work, after thorough deliberation, we have decided to withdraw our manuscript from consideration for publication in ICLR 2024.

Before proceeding with the withdrawal, we feel it is necessary to address a few critical points raised during the review. This is intended not as a defense for the sake of the current submission, but rather as a clarification for the academic community and for any future research we may conduct in this area. Our aim is to prevent any possible misunderstandings that might arise from the critiques and to contribute constructively to the ongoing dialogue in our field.

- The novelty of our proposed penetration loss term

    We proposed two neural physical losses for the intricate and non-differentiable process of verifying the plausibility of hand motions, one focusing on hand-object penetration. Though previous works have proposed some methods to mitigate penetration, we argue that our proposed penetration depth metric targets at the under-explored cases of deep penetration and thin structures, while existing loss terms tend to get stuck at local minima and fail to remedy this issue.

    With the proposed metric, we aim to address the problem of deep penetration, where the hand mesh completely penetrates through the manipulated object (typically at the thin and delicate parts of the object), e.g. the example shown in figure 2 of the main paper. This issue is common in noisy HOI data and greatly damages the data quality, but lacks satisfactory solutions. According to quantitive and qualitative evaluation, our proposed grasp credibility loss, which is trained with the PD metric, can remedy this issue effectively.

    Previous works mainly worked on the track of measuring how deep vertices of one mesh are inside another mesh to quantify the severity of interpenetration. While this approach can help with mild penetration issues by pulling vertices of one mesh out of another mesh, they perform poorly on deep penetration cases.

    When deep penetration happens, the direction to move the vertices to resolve penetration becomes ambiguous as the hand mesh lies on both side of the object, and it is very challenging to have a differentiable term producing correct gradients. While our method understands how to handle deep penetration through data prior learned from the penetration depth metric.

---

> ### Author Response · Authors · 2023-11-18
>
> - The setting of de-noising hand pose
>
>     We adopted the current setting as it has important research value and practical applications.
>
>     Beyond refining tracking results for traditional hand object interaction reconstruction tasks, our setting is also critical for broader applications such as virtual object manipulation and interaction retargeting in VR/AR and gaming, where clean object trajectory is usually accessible.
>
>     First, our method can be potentially used for manipulating virtual object by retargeting interaction. In this setting, the trajectory of the manipulated virtual object derives from the tracking result of handheld controllers, which typically ensures satisfactory accuracy, while the hand poses need to be refined. [AR1] includes survey on these applications. Besides, some other works achieve virtual object manipulation from the motion of bare hands. These works, such as [AR2], track the hand pose and estimate the “forces” exerted by the real hand to the virtual object by evaluating their “intersection”, and use physical engines and other tools to obtain smooth and plausible object trajectory. While the object motions are typically plausible, the tracked hand poses can have penetration and other artifacts with the object due to simplification and smoothing in the physical simulation process, and need to be further de-noised conditioning on the object trajectory. While [AR2] uses simple heuristic method to post-process the hand poses, our method can be applied to such scenario.
>
>     What’s more, we would like to point out that for marker-based hand object interaction tracking popular for AR/VR or games, tracking articulated rigid body with many joints like hands is much more challenging than tracking rigid objects. Due to hand’s high degree of freedom, self-similarity, heavy self-occlusion and subtle contact with the object, even for the most sophisticated motion capture system, the quality of the tracked object is higher than that of the tracked hand. A very recent sophisticated hand-object interaction capturing system [AR3] employed markers and 54 high-resolution cameras, yet the quality of tracked hand is still lower compared with object, with artifacts such as the penetration between the hand and the scissors in figure 5 of [AR3]’s supplementary material.
>
>     [AR1] Mendes, Daniel, et al. "A survey on 3d virtual object manipulation: From the desktop to immersive virtual environments." *Computer graphics forum*. Vol. 38. No. 1. 2019.
>
>     [AR2] Höll, Markus, et al. "Efficient physics-based implementation for realistic hand-object interaction in virtual reality." *2018 IEEE conference on virtual reality and 3D user interfaces (VR)*. IEEE, 2018.
>
>     [AR3] Fan, Zicong, et al. "ARCTIC: A Dataset for Dexterous Bimanual Hand-Object Manipulation." *Proceedings of the IEEE/CVF Conference on Computer Vision and Pattern Recognition*. 2023.
>
>     Indeed, optimizing noisy object and hand movement simultaneously is an exciting and ambitious goal. Yet our setting is already challenging and significant considering the non-triviality posed by the dexterity of in-hand manipulation, the ambiguity of fixing deep penetration cases, etc. We believe our proposed method is a significant step for the research society to approach further goals.

---

> > ### Author Response · Authors · 2023-11-18
> >
> > - Physical basis and empirical settings in the two losses
> >
> >     We base our physical loss terms on some assumption on physical properties, such as object mass and the frictional coefficient. We hereby justify that these assumptions come with little cost.
> >
> >     Essentially, telling whether the given hand poses can feasibly manipulate the object along its trajectory doesn’t require the absolute values of forces exerted at each contact point, and it’s actually practically impossible to obtain the absolute values without sensors due to force ambiguity, i.e., different force combinations can have the same result. The force calculation process in the two metrics is more of an intermediate means for evaluating whether the given hand pose can possibly supply force in a certain direction (in force error), and if not, how far it is from the closest plausible hand pose (in manipulation expense). Our ultimate goal is to train a neural loss that differentiably quantifies the manipulation feasibility, with the knowledge of the two metrics.
> >
> >     - Setting the mass to a fixed value will not influence our de-noising process since the object mass is only a relative value for solving forces, and doesn’t affect resultant force distribution of the optimization process. Different mass values will result in the same force error since this metric reflects the relative error. The manipulation expense would be scaled proportional to the mass, but as long as we use the same mass value for the whole dataset, the relative values of manipulation expense can still reflect the plausibility of different hand poses.
> >     - Setting a fixed friction coefficient is a common practice in previous works [AR1], [AR2] though it could theoretically influence the denoising process (for example if we set the friction coefficient to zero it would be very hard to support the manipulation anymore).
> >
> >         In practice, we find our method to be quite robust. Our early observation reveals that as long as the selected friction coefficient isn’t too off, the result of force error and manipulation expense can align well with human perception concerning the manipulation feasibility.
> >
> >         A better solution might be training neural losses with different friction coefficients and using system identification to adapt the method to different objects, yet that would be out of this work’s scope. We will add this discussion to make the assumption clearer.
> >
> >
> >     [AR1] Hu, Haoyu, et al. "Physical interaction: Reconstructing hand-object interactions with physics." *SIGGRAPH Asia 2022 Conference Papers*. 2022.
> >
> >     [AR2] Zhang, He, et al. "Manipnet: neural manipulation synthesis with a hand-object spatial representation." *ACM Transactions on Graphics (ToG)* 40.4 (2021): 1-14.